# Impact of high body mass index on hepatocellular carcinoma risk in chronic liver disease: A population-based prospective cohort study

**Moonho Kim**[1‡], **Baek Gyu Jun**[2‡], **Hwang Sik Shin**[3], **Jee-Jeon Yi**[4], **Sang Gyune Kim**[5]*, **Sang-Wook Yi**[6]*

1 Department of Hematology and Oncology, University of Ulsan College of Medicine, Gangneung Asan Hospital, Gangneung, South Korea, 2 Department of Internal Medicine, Asanseoul Internal Medicine Clinic, Seoul, South Korea, 3 Department of Family Medicine, Soonchunhyang University Hospital Cheonan, Soonchunhyang University College of Medicine, Cheonan, South Korea, 4 Institute for Occupational and Environmental Health, Catholic Kwandong University, Gangneung, South Korea, 5 Department of Internal Medicine, Soonchunhyang University College of Medicine Bucheon Hospital, Bucheon, South Korea, 6 Department of Preventive Medicine and Public Health, Catholic Kwandong University College of Medicine, Gangneung, South Korea

‡ MK and BGJ shared co-first authorship on this work.
* mcnulty@schmc.ac.kr (SGK); flyhigh@cku.ac.kr (SWY)

**Data Availability Statement:** The data that support the findings of this study are available from the National Health Insurance Service (NHIS) [http://nhiss.nhis.or.kr/bd/ab/bdaba000eng.do], but

## Abstract

### Background and aims

We investigated associations between body mass index (BMI) and hepatocellular carcinoma (HCC) in patients with hepatitis B (HBV) C (HCV) virus infection, alcoholic liver disease (ALD), non-alcoholic fatty liver disease (NAFLD), and liver cirrhosis (LC).

### Methods

We followed 350,608 Korean patients with liver disease who underwent routine health examinations from 2003–2006 until December 2018 via national hospital discharge records. Multivariable adjusted hazard ratios (HRs) per 5-kg/m² BMI increase (BMI ≥25 kg/m²) for HCC risk were calculated using Cox models. HCC developed in 17,752 patients.

### Results

The HRs (95% CI) were 1.17 (1.06–1.28), 1.08 (0.87–1.34), 1.34 (1.14–1.58), 1.51 (1.17–1.94), and 1.11 (1.00–1.23) for HBV, HCV, ALD, NAFLD, and LC, respectively. The HRs for HBV were 1.45 (1.23–1.70) and 1.06 (0.95–1.19) in women and men, respectively; the corresponding HRs for LC were 1.27 (1.07–1.50) and 1.02 (0.90–1.16), respectively. In patients <65 years old with HBV, HCV, and NAFLD, the HRs were 1.17 (1.07–1.29), 1.33 (1.03–1.73), and 1.20 (0.87–1.64), respectively; the corresponding HRs were 1.05 (0.70–1.59), 0.74 (0.50–1.10), and 2.40 (1.62–3.54), respectively, in patients ≥65 years old. A BMI of 27.5–29.9 kg/m² showed significantly higher HCC risks in patients with HBV, ALD, NAFLD, and LC.

restrictions apply to the availability of these data, which were used under license for the current study, and so are not publicly available.

**Funding:** This study was supported by The Research Supporting Program of The Korean Association for the Study of the Liver and The Korean Liver Foundation (KASLKLF2021-08) and the Medical Research Promotion Program through Gangneung Asan Hospital funded by the Asan Foundation (2022II0016).

**Competing interests:** The authors have declared that no competing interests exist.

## Conclusions

Higher BMIs were associated with increased HCC risks in patients with HBV, ALD, NAFLD, and LC. Overweight status increased HCC risk. Women with HBV and LC had stronger BMI-HCC associations than men. The effect of high BMI was stronger in older patients with NAFLD and younger patients with viral hepatitis.

## Introduction

Hepatitis B virus (HBV) infection, hepatitis C virus (HCV) infection, and liver cirrhosis (LC) are well-known risk factors for hepatocellular carcinoma (HCC). The association between obesity and HCC in patients with these high-risk liver diseases (HLDs) is not well understood, and the role of a high body mass index (BMI) in HCC development and progression has not been clearly elucidated. Some studies have shown a positive association between BMI and HCC risk [1–5], whereas others have not [6, 7]. In patients with less severe liver diseases, such as non-alcoholic fatty liver disease (NAFLD) without LC, few studies have examined these associations; hence, the associations remain unclear. For example, in patients with NAFLD, obesity is not associated with a higher HCC risk [8].

We aimed to investigate the BMI-HCC association in patients with various liver diseases (HBV infection, HCV infection, LC, NAFLD, and alcoholic liver disease [ALD]). In addition, as previous studies have suggested potentially different BMI-HCC associations between men and women and between younger and older adults, we further investigated sex- and age-specific associations [6, 9–12].

## Patients and methods

### Study population and follow-up

This population-based prospective cohort study enrolled 355,670 Koreans with liver diseases (HBV infection, HCV infection, LC, ALD, and NAFLD) aged 18–99 years who were examined between 2003 and 2006 and had no known cancer or missing information on variables and examination date. Baseline data were collected at the time of the health examination (index date). We excluded 5,062 patients with multiple liver diseases. We followed the remaining 350,608 included patients (Fig 1) until December 31, 2018, via record linkage to hospital discharge records from the National Health Insurance Service (NHIS), in which certified health information managers reviewed medical records and assigned standardized diagnosis codes. All patients discharged from the hospital due to HCC (International Classification of Diseases 10th Revision code C220) for the first time were considered incident cases. The authors were granted access to anonymized data from the NHIS (From June 1, 2022, to June 30, 2023). This study was approved by the Institutional Review Board of Gangneung Asan Hospital, Gangneung, Republic of Korea (GNAH 2022-04-005). Informed consent was waived owing to the use of anonymized data that were constructed and provided by the NHIS according to a strict confidentiality protocol. All research was conducted in accordance with both the Declarations of Helsinki and Istanbul.

### Completeness of HCC incidence data by the NHIS

Ninety-seven percent of Koreans are NHIS-insured [13]. Additionally, patients with HLDs underwent liver cancer surveillance twice a year. Patients with high alpha-fetoprotein levels or a suspicious mass on ultrasonography were referred for HCC diagnosis using dynamic

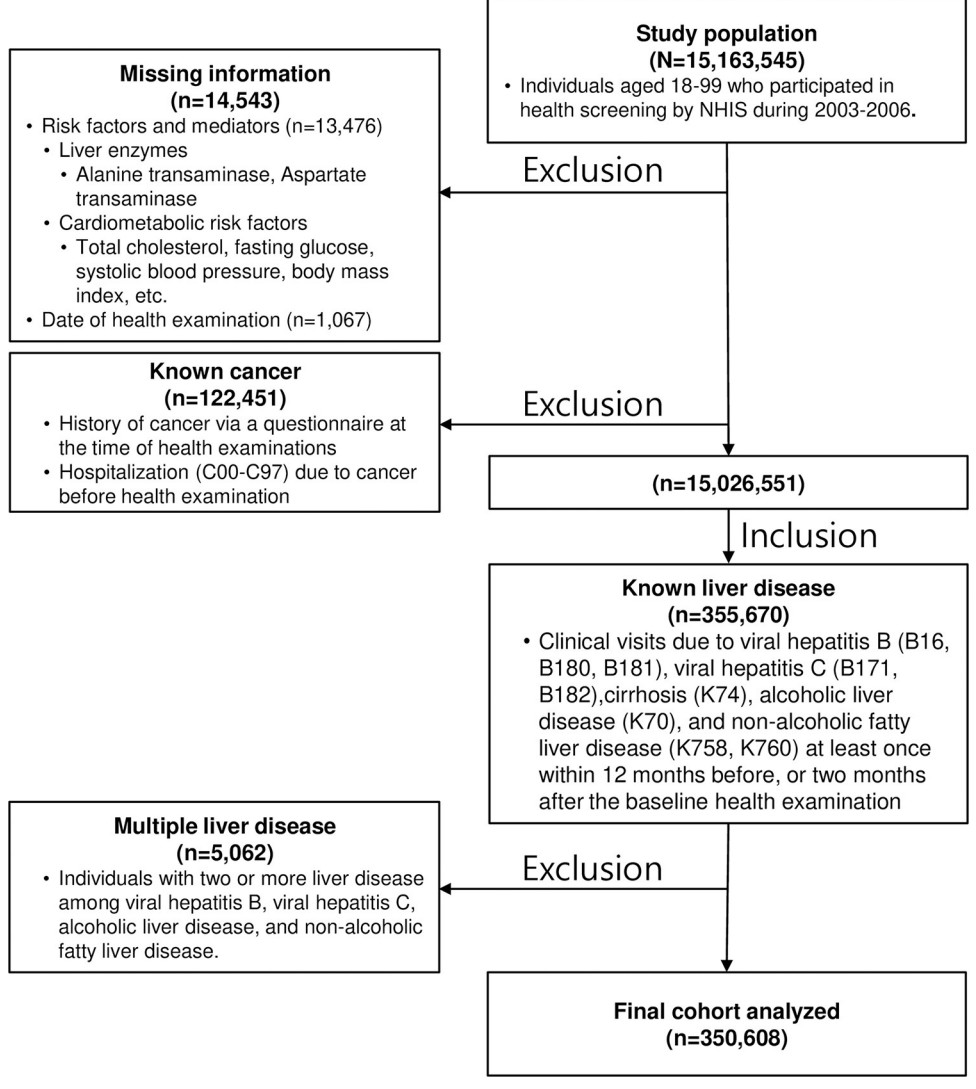

**Fig 1. Flow diagram of the study cohort.**

computed tomography or magnetic resonance imaging. All the procedures were covered by the NHIS. The completeness of cancer incidence data from the NHIS is comparable to that of the Korea National Cancer Incidence Database (> 95% for liver cancer) [14, 15].

## Data collection

Data were collected during baseline examinations using measurements and questionnaires. Alanine transaminase and aspartate transaminase levels were measured using the

nicotinamide adenine dinucleotide-ultraviolet or Reitman-Frankel method. Fasting serum glucose and total cholesterol levels were assessed using enzymatic methods [16]. Blood pressure was measured using a standard mercury sphygmomanometer. BMI was calculated as the measured weight (kg) divided by the square of measured height ($m^2$). Smoking status, alcohol use, and history of cancer and cardiovascular disease were assessed using a questionnaire. Patients with a self-reported cancer history or patients admitted to a hospital for cancer before the baseline examination were considered to have preexisting cancer. Patient examinations and data collection followed a standard protocol documented by the government. The data collection methods for smoking and alcohol consumption were similar to those used in our previous study [7].

## Prevalent diseases at baseline

We considered patients to have a baseline prevalent disease if they visited a hospital for the disease at least once within 12 months before or 2 months after the baseline examination. The diseases were selected using International Classification of Diseases 10th Revision codes: HBV infection (B16, B180, and B181), HCV infection (B171 and B182), diabetes (E10-E14), ALD (K70), LC (K74), and NAFLD (K758 and K760).

## Statistical analysis

BMI was categorized into seven groups: <18.5, 18.5–20.9, 21–22.9, 23–24.9, 25–27.4, 27.5–29.9, and ≥30 kg/$m^2$. BMI was also analyzed as a continuous variable (per 5 kg/$m^2$ increase), assuming a linear association in the full, lower (<25 kg/$m^2$), and upper (≥25 kg/$m^2$) ranges. The effects of BMI on HCC were evaluated using stratified analysis across liver disease status. In the subgroup analysis, the BMI-HCC associations in patients with LC were examined according to LC etiology (HBV, HCV, ALD, or NAFLD).

The HRs for HCC incidence were obtained using Cox proportional hazards models stratified by age (years) at baseline (18–24, 25–34, 35–44, 45–54, 55–64, 65–74, 75–84, and 85–99 years, using the STRATA statement). The multivariable analysis was adjusted for age at baseline (continuous variable), sex, smoking status (never, former, or current smoker [<10, 10–19, or ≥20 cigarettes/day]), alcohol use (none, <10, 10–19, 20–39, and ≥40 g of ethanol/day; or missing information), physical activity (exercise with light sweating, none, 1–2 times/week, and 3–7 times/week), and income status (quartiles; 1 [low income], 2, 3, 4 [high income]). BMI effect mediators such as glycemic status (normoglycemia [<100 mg/dL], impaired fasting glucose levels [100–125 mg/dL], diabetes [≥126 mg/dL or prevalent diabetes]), total cholesterol levels (continuous variable) [17], and alanine transaminase levels (natural log-transformed levels) were further adjusted for in sensitivity analyses. Sex- and age-stratified analyses were performed.

Effect size differences between the sexes and age groups were evaluated using the Cochrane Q statistics test as an interaction test. All *P*-values were two-sided, and the analyses were performed using SAS version 9.4 (SAS Institute Inc., Cary, NC, USA).

## Results

### Baseline characteristics

HBV, HCV, and LC were present in 120,994, 14,882, and 31,260 patients, respectively (Table 1). During a mean follow-up period of 13.7 years, HCC was diagnosed in 8,492 patients. The proportions of patients with HBV infection, HCV infection, and LC having a BMI ≥25 kg/$m^2$ were 32.5%, 35.4%, and 34%, respectively. Patients with HBV infection were younger

**Table 1. Baseline demographic and clinical characteristics.**

| Characteristic | HBV n = 120,994 | HCV n = 14,882 | ALD n = 106,112 | NAFLD n = 88,353 | LC n = 31,260 |
|---|---|---|---|---|---|
| HCC | 9,043 | 1,654 | 2,659 | 650 | 6,506 |
| Sex | | | | | |
| Men | 76,067 (62.9) | 8,274 (55.6) | 95,767 (90.3) | 53,870 (61.0) | 22,316 (71.4) |
| Women | 44,927 (37.1) | 6,608 (44.4) | 10,345 (9.7) | 34,483 (39.0) | 8,944 (28.6) |
| BMI, kg/m$^2$ | | | | | |
| <18.5 | 3,824 (3.2) | 383 (2.6) | 2,817 (2.7) | 1,303 (1.5) | 933 (3.0) |
| 18.5–20.9 | 18,815 (15.6) | 1,889 (12.7) | 12,981 (12.2) | 7,001 (7.9) | 4,384 (14.0) |
| 21–22.9 | 27,659 (22.9) | 3,190 (21.4) | 20,412 (19.2) | 14,422 (16.3) | 7,133 (22.8) |
| 23–24.9 | 31,441 (26.0) | 4,150 (27.9) | 27,028 (25.5) | 22,751 (25.8) | 8,184 (26.2) |
| 25–27.4 | 26,392 (21.8) | 3,466 (23.3) | 27,419 (25.8) | 25,432 (28.8) | 6,987 (22.4) |
| 27.5–29.9 | 9,389 (7.8) | 1,302 (8.7) | 11,257 (10.6) | 11,995 (13.6) | 2,668 (8.5) |
| ≥30 | 3,474 (2.9) | 502 (3.4) | 4,198 (4.0) | 5,449 (6.2) | 971 (3.1) |
| Glycemic status | | | | | |
| Normoglycemia | 87,980 (72.7) | 9,505 (63.9) | 61,990 (58.4) | 56,414 (63.9) | 18,449 (59.0) |
| IFG | 23,302 (19.3) | 3,278 (22.0) | 28,530 (26.9) | 21,181 (24.0) | 6,855 (21.9) |
| Diabetes | 9,712 (8.0) | 2,099 (14.1) | 15,592 (14.7) | 10,758 (12.2) | 5,956 (19.1) |
| Smoking, pack/day | | | | | |
| Never | 76,565 (63.3) | 10,087 (67.8) | 302 (0.3) | 157 (0.2) | 19,013 (60.8) |
| Former | 12,537 (10.4) | 1,447 (9.7) | 42,255 (39.8) | 55,046 (62.3) | 3,430 (11.0) |
| <0.5 | 7,777 (6.4) | 872 (5.9) | 15,196 (14.3) | 9,758 (11.0) | 2,765 (8.8) |
| 0.5–0.9 | 15,999 (13.2) | 1,494 (10.0) | 9,684 (9.1) | 4,885 (5.5) | 3,857 (12.3) |
| 1–1.9 | 5,267 (4.4) | 592 (4.0) | 24,181 (22.8) | 11,196 (12.7) | 1,394 (4.5) |
| ≥2 | 2,634 (2.2) | 373 (2.5) | 12,818 (12.1) | 5,436 (6.2) | 737 (2.4) |
| Unknown | 215 (0.2) | 17 (0.1) | 1,676 (1.6) | 1,875 (2.1) | 64 (0.2) |
| Alcohol, ethanol (g)/day | | | | | |
| None | 68,917 (57.0) | 9,676 (65.0) | 1,833 (1.7) | 1,739 (2.0) | 20,007 (64.0) |
| <10 | 27,974 (23.1) | 2,581 (17.3) | 28,734 (27.1) | 46,244 (52.3) | 4,763 (15.2) |
| 10–19 | 13,670 (11.3) | 1,330 (8.9) | 21,144 (19.9) | 18,797 (21.3) | 2,561 (8.2) |
| 20–39 | 3,908 (3.2) | 399 (2.7) | 22,447 (21.2) | 12,014 (13.6) | 1,078 (3.4) |
| ≥40 | 3,972 (3.3) | 492 (3.3) | 11,479 (10.8) | 4,378 (5.0) | 2,039 (6.5) |
| Unknown | 2,553 (2.1) | 404 (2.7) | 20,475 (19.3) | 5,181 (5.9) | 812 (2.6) |
| Physical activity, times/week | | | | | |
| None | 62,244 (51.4) | 7,994 (53.7) | 59,009 (55.6) | 46,538 (52.7) | 18,044 (57.7) |
| 1–2 | 34,625 (28.6) | 3,532 (23.7) | 27,337 (25.8) | 23,294 (26.4) | 7,137 (22.8) |
| ≥3 | 24,125 (19.9) | 3,356 (22.6) | 19,766 (18.6) | 18,521 (21.0) | 6,079 (19.4) |
| Income status, quartile | | | | | |
| Q1 (low) | 24,235 (20.0) | 2,788 (18.7) | 20,164 (19.0) | 16,790 (19.0) | 6,076 (19.4) |
| Q2 | 24,333 (20.1) | 2,689 (18.1) | 22,816 (21.5) | 16,654 (18.8) | 5,914 (18.9) |
| Q3 | 31,972 (26.4) | 3,866 (26.0) | 30,046 (28.3) | 23,338 (26.4) | 7,890 (25.2) |
| Q4 | 40,454 (33.4) | 5,539 (37.2) | 33,086 (31.2) | 31,571 (35.7) | 11,380 (36.4) |
| Age groups, years | | | | | |
| <65 | 11,4728 (94.8) | 12,008 (80.7) | 93,860 (88.5) | 79,551 (90.0) | 25,686 (82.2) |
| ≥65 | 6,266 (5.2) | 2,874 (19.3) | 12,252 (11.5) | 8,802 (10.0) | 5,574 (17.8) |
| Total cholesterol, mg/dL | | | | | |
| <200 | 79,873 (66.0) | 10,385 (69.8) | 59,626 (56.2) | 43,981 (49.8) | 23,348 (74.7) |
| 200–239 | 31,497 (26.0) | 3,401 (22.9) | 32,149 (30.3) | 30,222 (34.2) | 5,975 (19.1) |

*(Continued)*

**Table 1.** (Continued)

| Characteristic | HBV n = 120,994 | HCV n = 14,882 | ALD n = 106,112 | NAFLD n = 88,353 | LC n = 31,260 |
|---|---|---|---|---|---|
| ≥240 | 9,624 (8.0) | 1,096 (7.4) | 14,337 (13.5) | 14,150 (16.0) | 1,937 (6.2) |

Data are expressed as numbers and percentages.

Abbreviations: ALD, alcoholic liver disease; LC, liver cirrhosis; HCC, hepatocellular carcinoma; BMI, body mass index; IFG, impaired fasting glucose

and had higher total cholesterol levels. Patients with LC were predominantly men and smokers, with higher glucose levels, more alcohol use, and less physical activity. HBV was the most common etiology of LC, followed by ALD (S1 Table).

## Categorical analyses

HRs for HCC were compared based on a BMI range of 23–24.9 kg/m$^2$. In patients with HBV infection, the HRs of HCC were increased in the BMI ranges of 27.5–29.9 (HR 1.09, 95% CI 1.01–1.18) and ≥30 kg/m$^2$ (HR 1.20, 95% CI 1.06–1.36) (Fig 2; S2 Table). In patients with HCV, the HRs were 0.99 (95% CI 0.86–1.13), 1.09 (95% CI 0.91–1.31), and 1.00 (95% CI 0.73–1.36) for BMI ranges of 25–27.4, 27.5–29.9, and ≥30 kg/m$^2$, respectively. In patients with ALD, the HRs of HCC were significantly increased in the BMI ranges of 27.5–29.9 (HR 1.22, 95% CI 1.06–1.42) and ≥30 kg/m$^2$ (HR 1.38, 95% CI 1.10–1.74). Similarly, in patients with NAFLD, the HRs of HCC were increased in the BMI ranges of 27.5–29.9 (HR 1.36, 95% CI 1.06–1.74) and ≥30 kg/m$^2$ (HR 1.42, 95% CI 1.00–2.01). In patients with LC, the HRs of HCC were 1.00 (95% CI 0.93–1.07), 1.11 (95% CI 1.02–1.22), and 1.07 (95% CI 0.93–1.24) in BMI ranges of 25–27.4, 27.5–29.9, and ≥30 kg/m$^2$, respectively. In the subgroup analysis of LC etiology, patients with HBV-LC co-occurrence showed increased HCC risks in the BMI range of 27.5–29.9 kg/m$^2$ (HR 1.35, 95% CI 1.15–1.59).

## Linear analyses

In linear analyses according to BMI, for each 5 kg/m$^2$ increase from a BMI ≥25 kg/m$^2$, the multivariable-adjusted HRs were 1.17 ($P = 0.001$), 1.08 ($P = 0.486$), 1.11 ($P = 0.047$), 1.34 ($P<0.001$), and 1.51 ($P = 0.001$) for HBV infection, HCV infection, LC, ALD, and NAFLD, respectively (Fig 2). Overall, positive BMI-HCC associations were shown in patients with chronic liver disease (CLD). In the subgroup analysis for patients with LC (BMI ≥25 kg/m$^2$), the HRs for HCC per 5 kg/m$^2$ increase in BMI were 1.19 (0.98–1.45, $P = 0.075$), 1.38 (0.75–2.51, $P = 0.297$), 1.36 (0.92–2.02, $P = 0.125$), and 1.59 (0.53–4.76, $P = 0.410$) in patients with HBV infection, HCV infection, ALD, and NAFLD, respectively (S3 Table).

## Age- and sex-stratified analyses

In the sex-stratified analyses (BMI ≥25 kg/m$^2$), women with HBV infection (1.45 vs. 1.06, $P$ for interaction = 0.002) and LC (1.27 vs. 1.02, $P$ for interaction = 0.050) showed stronger positive BMI-HCC associations than in men (Fig 3). However, for patients with NAFLD, the association was stronger in men than in women (1.84 vs. 1.03, $P$ for interaction = 0.039). The association showed no sex difference in patients with ALD (1.35 vs. 1.30, $P$ for interaction = 0.913). In the age-stratified analyses (patients divided into groups aged <65 years and ≥65 years; BMI ≥25 kg/m$^2$), younger patients with HBV infection (1.17 vs. 1.05, $P$ for interaction = 0.606) and HCV infection (1.33 vs. 0.77, $P$ for interaction = 0.015) had a higher HCC risk than older patients (Fig 3). In the subgroup analysis of patients with LC, women with

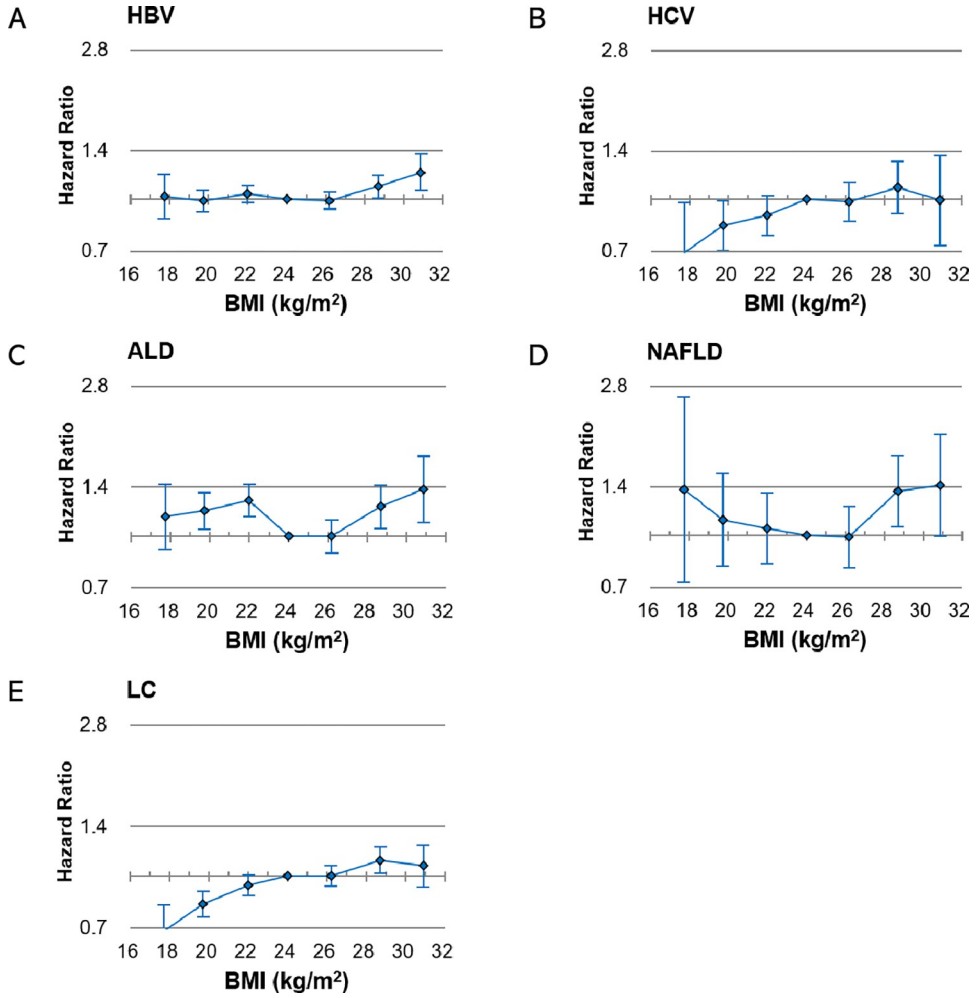

**Fig 2. Hepatocellular carcinoma risk according to liver disease, for each 5-kg/m$^2$ BMI increase in patients with BMI ≥25 kg/m$^2$.** HBV, hepatitis B virus; HCV, hepatitis C virus; ALD, alcoholic liver disease; NAFLD, nonalcoholic fatty liver disease; LC, liver cirrhosis; BMI, body mass index.

HBV infection (1.73 vs. 0.95, *P* for interaction = 0.003) and older patients with NAFLD (11.08 vs. 0.76, *P* for interaction = 0.036) had significantly higher risks of HCC associated with a higher BMI than men with HBV and younger patients with NAFLD, respectively (S4 Table).

## Discussion

This prospective cohort study of more than 350,000 patients with CLDs showed that obesity increased HCC risk. In overweight (BMI ≥25 and <30 kg/m$^2$) and obese (BMI ≥30 kg/m$^2$) patients with HBV infection, LC, ALD, or NALFD, BMI increase was associated with higher HCC risks. In patients with a BMI ≥25 kg/m$^2$, HCC risks associated with higher BMIs were more prominent in women than in men with HBV infection, HCV infection, or LC. Age-specific BMI-HCC associations differed by CLDs; the associations were stronger in younger adults than in older adults with HBV and HCV infections but stronger in older adults than in younger adults with NAFLD.

In our study, obesity increased the HCC risk in patients with HBV infection. Some previous studies have shown inconsistent associations between obesity and HCC risk in Western and

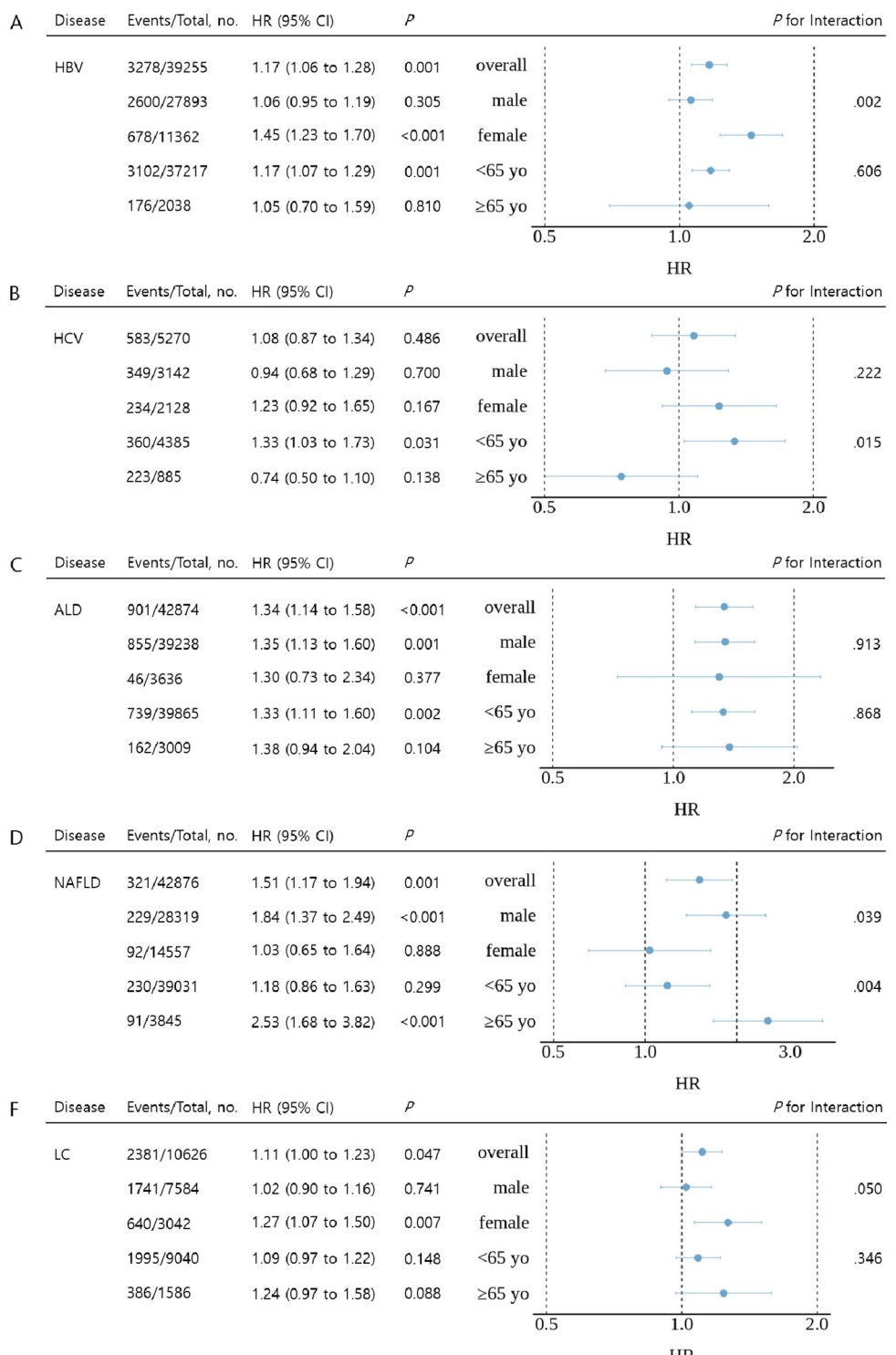

**Fig 3. Cox regression analysis of hepatocellular carcinoma risk according to liver disease, for each 5-kg/m$^2$ BMI increase in patients with BMI $\geq$25 kg/m$^2$.** HBV, hepatitis B virus; HCV, hepatitis C virus; ALD, alcoholic liver disease; NAFLD, nonalcoholic fatty liver disease; LC, liver cirrhosis; BMI, body mass index.

Taiwanese patients with HBV infection [3, 18, 19], although the association was consistent in other studies [4, 20–22]. Recently, a large-scale Korean study reported an association between a high BMI and HCC risk [23]. Our study confirmed that obesity increased HCC incidence in Koreans with HBV infection. Previous studies may not have found a significant BMI-HCC association, mainly because of the relatively small study sample sizes and the modest nature of true associations. Notably, our study showed that overweight status (especially a BMI of 27.5–29.9 kg/m$^2$) was associated with higher HCC risks.

In our study, patients with HCV infection showed a modest obesity-HCC association. Some previous studies showed increased obesity-associated HCC risks [4], although others did not [24–26]. In the current study, higher BMIs were significantly associated with HCC risk in patients with HBV infection but not in patients with HCV infection. Associations related to HCV than HBV were weaker in HBV-endemic areas [7], whereas the opposite findings have been reported in other areas. These findings might be partially explained by the fact that antiHCV medication efficacy was lower than anti-HBV medication efficacy in the early 2000s. The formal heterogeneity test showed no difference in the BMI-HCC association between HBV and HCV infections (*P* for heterogeneity = 0.509) in the current study. The modest association in patients with HCV infection was not significant because of the relatively small number of patients with HCV infection compared with the number of patients with HBV infection in the present study.

In previous studies, obesity was associated with HCC development, mostly in patients in Western countries with ALD-LC and HCV-LC co-occurrences [1, 5, 24, 27, 28]. The HCC risk associated with obesity was more pronounced in patients with ALD-LC co-occurrence than in patients with viral hepatitis-LC co-occurrence [5, 24]. In the current study, the BMI-HCC association was stronger in patients with ALD-LC and HCV-LC co-occurrences than in patients with HBV-LC co-occurrence. However, the association was not significant owing to the smaller number of participants with HCV infection.

In our study, NAFLD patients with overweight and obesity showed an increased HCC risk; moreover, a 5-kg/m$^2$ BMI increment increased HCC risk by 51%. NAFLD is a known risk factor for HCC, and a BMI increase is associated with a higher NAFLD prevalence; nevertheless, few studies have reported an obesity-related increase in HCC risk in patients with NAFLD [24, 27, 29, 30]. The absence of a BMI-HCC association may be explained, at least partially, by crude BMI classification into two groups ([≥30 vs. <30 kg/m$^2$] [8, 29] and [≥25 vs. <25 kg/m$^2$] [30]), a retrospective study design [8, 24, 27], including patients with cirrhosis only [24, 27], and adjustment for the effect modifiers of BMI [24, 27]. We may have identified this association because of the prospective study design, the inclusion of patients with and without cirrhosis, the detailed BMI classification into seven groups, and the lack of adjustment for effect modifiers of BMI.

Regarding ALD, previous studies were conducted mostly in patients with cirrhosis, and few studies examined the BMI-HCC association in patients without LC [5, 27]. In the current study, patients with ALD without LC showed higher HCC risks associated with overweight and obesity, probably because of the synergistic interaction between alcohol intake and BMI [31]. The associations were similar in both sexes, albeit statistically insignificant in women due to the small number of women with ALD.

Furthermore, our study showed a higher relative risk of HCC associated with BMI in women than men with viral hepatitis-LC co-occurrence. Few studies have examined sex-specific BMI-HCC associations in patients with viral hepatitis-LC co-occurrence [6, 10, 23, 32–35]. In the current study, the BMI-HCC associations were similar between men and women with ALD; nevertheless, potentially stronger associations were observed in men than in women with NAFLD. Our study suggests different associations according to sex for each CLD.

Unfortunately, except for the presumption of sex-specific differences in the hepatocarcinogenic mechanisms of high BMIs and each liver disease, proposing a mechanism that can adequately explain the sex differences in the obesity-related HCC risk observed in our study is difficult. Further studies are needed to clarify the potentially different associations according to sex for each CLD.

In the age-specific analysis, relative HCC risks associated with a higher BMI were higher in younger (<65 years) than in older adults with HBV and HCV infections, whereas the risks were higher in older than in younger adults with NAFLD. Younger patients with viral hepatitis may experience a stronger synergistic effect between viral infection and obesity due to lower comorbidity rates [13, 36]. Regarding NAFLD, our finding of stronger associations in older adults may reflect the fact that high BMI affects HCC development in the long run. For HCC prevention, comorbid obesity may not be ignored in younger patients with viral hepatitis; however, more attention may be needed in NAFLD patients with obesity, especially in patients aged ≥65 years.

The role of obesity in hepatocarcinogenesis in patients with HLD is not well understood. Patients with HBV/HCV infections and LC have a 10–100-fold higher HCC risk than patients without HLDs, whereas obesity is associated with a 2–3-fold higher HCC risk in the general population. If HLDs and obesity are independent risk factors for HCC, the effect of obesity on HCC would be negligible. The current study showed that a higher BMI increased HCC risk in patients with CLDs, including HLDs. Our results indicate a strong synergistic interaction between the hepatocarcinogenic mechanisms of CLDs (including HLDs) and obesity [37]. Detailed mechanisms underlying these synergistic interactions are yet to be elucidated.

This study had some strengths. We examined the relationship between HCC risk and obesity in patients with major liver diseases, such as HBV infection, HCV infection, LC, and NAFLD. Previous studies examined the relationship between obesity and HCC risk in patients with a single liver disease. However, we analyzed HCC risk in patients with each type of HLD in a nationwide cohort. Furthermore, this was a large-scale study that analyzed the relationship between HCC risk and obesity by age and sex. We also tested our hypothesis by adjusting for the main confounding variables but not for the effect mediators of obesity [38]. We used a prospective cohort design to minimize recall and selection biases related to retrospective studies.

Our study had several limitations. First, obesity was defined in terms of BMI only; other factors, such as visceral obesity (evaluated using the waist-to-hip ratio), could not be investigated owing to the nature of the raw data. Second, important data such as fibrosis score, antiviral therapies, and detailed viral factors for HBV and HCV were not available. For example, antiviral therapies are known to reduce the risk of HBV- and HCV-related HCC. Third, the proportion of Koreans with a BMI >30 kg/m$^2$ (especially >35 kg/m$^2$) was small. Therefore, the relationship between obesity and HCC incidence may have been underestimated. Fourth, we could not analyze the impact of the change in BMI on HCC risk. Only a few studies examined the association between BMI (or weight change) and HCC in persons with liver diseases [39]. The results were inconsistent. Our analysis based on a single measurement of BMI may underestimate the true association of dynamic change in BMI. Finally, the study population included only Koreans, which may limit its interpretation and applicability. For example, most cases of LC in Korea are caused by HBV infection. The potential differences in associations across LC etiologies suggest that the BMI-HCC association in patients with LC may depend on the population-specific distribution of the LC etiology.

In conclusion, in patients with CLD, obesity and overweight (specifically a BMI of 27.5–29.9 kg/m$^2$) were associated with an increased HCC risk. In overweight and obese patients with CLDs, a BMI increase was associated with a higher HCC risk. The risk of HCC associated with a higher BMI was more pronounced in women than in men with viral hepatitis and LC.

In patients with NAFLD, the impact of a high BMI on HCC was greater in older patients ($\geq$65 years).

## Supporting information

**S1 Table. Patients' baseline demographic and clinical characteristics.**
(DOCX)

**S2 Table. Association between BMI and hepatocellular carcinoma (HCC) risk by liver disease.**
(DOCX)

**S3 Table. Association between BMI and HCC risk by liver disease etiology, for each 5-kg/m$^2$ BMI increase (BMI $\geq$25 kg/m$^2$).**
(DOCX)

**S4 Table. Cox regression of HCC risk by liver cirrhosis etiology, per 5-kg/m$^2$ BMI ($\geq$25 kg/m$^2$) increase.**
(DOCX)

## Acknowledgments

This research was supported by data from the National Health Insurance Service of Korea (NHIS-2024-1-264).

## Author Contributions

**Conceptualization:** Sang Gyune Kim, Sang-Wook Yi.

**Data curation:** Jee-Jeon Yi, Sang-Wook Yi.

**Formal analysis:** Moonho Kim, Baek Gyu Jun, Hwang Sik Shin, Sang Gyune Kim, Sang-Wook Yi.

**Funding acquisition:** Moonho Kim, Baek Gyu Jun, Sang Gyune Kim.

**Investigation:** Baek Gyu Jun, Hwang Sik Shin, Jee-Jeon Yi.

**Methodology:** Sang Gyune Kim, Sang-Wook Yi.

**Project administration:** Sang-Wook Yi.

**Resources:** Moonho Kim, Jee-Jeon Yi, Sang-Wook Yi.

**Supervision:** Sang-Wook Yi.

**Writing – original draft:** Moonho Kim, Baek Gyu Jun.

**Writing – review & editing:** Moonho Kim, Baek Gyu Jun, Hwang Sik Shin, Jee-Jeon Yi, Sang Gyune Kim, Sang-Wook Yi.

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
