## [Decision Letter · Decision Letter 0]

4 Sep 2024

PONE-D-24-15224Impact of high body mass index on hepatocellular carcinoma risk in chronic liver disease: A population-based prospective cohort studyPLOS ONE

Dear Dr. Yi,

Thank you for submitting your manuscript to PLOS ONE. After careful consideration, we feel that it has merit but does not fully meet PLOS ONE’s publication criteria as it currently stands. Therefore, we invite you to submit a revised version of the manuscript that addresses the points raised during the review process.

I would like to sincerely apologise for the delay you have incurred with your submission. It has been exceptionally difficult to secure reviewers to evaluate your study. We have now received two completed reviews; the comments are available below. The reviewers have raised significant scientific concerns about the study that need to be addressed in a revision.

Please revise the manuscript to address all the reviewer's comments in a point-by-point response in order to ensure it is meeting the journal's publication criteria. Please note that the revised manuscript will need to undergo further review, we thus cannot at this point anticipate the outcome of the evaluation process.

We look forward to receiving your revised manuscript.

Kind regards,

Miquel Vall-llosera Camps

Senior Staff Editor

PLOS ONE

Journal Requirements:

1. When submitting your revision, we need you to address these additional requirements. Please ensure that your manuscript meets PLOS ONE's style requirements, including those for file naming. The PLOS ONE style templates can be found at https://journals.plos.org/plosone/s/file?id=wjVg/PLOSOne_formatting_sample_main_body.pdf and https://journals.plos.org/plosone/s/file?id=ba62/PLOSOne_formatting_sample_title_authors_affiliations.pdf 2. We note that you have indicated that there are restrictions to data sharing for this study. For studies involving human research participant data or other sensitive data, we encourage authors to share de-identified or anonymized data. However, when data cannot be publicly shared for ethical reasons, we allow authors to make their data sets available upon request. For information on unacceptable data access restrictions, please see http://journals.plos.org/plosone/s/data-availability#loc-unacceptable-data-access-restrictions.  Before we proceed with your manuscript, please address the following prompts: a) If there are ethical or legal restrictions on sharing a de-identified data set, please explain them in detail (e.g., data contain potentially identifying or sensitive patient information, data are owned by a third-party organization, etc.) and who has imposed them (e.g., a Research Ethics Committee or Institutional Review Board, etc.). Please also provide contact information for a data access committee, ethics committee, or other institutional body to which data requests may be sent. b) If there are no restrictions, please upload the minimal anonymized data set necessary to replicate your study findings to a stable, public repository and provide us with the relevant URLs, DOIs, or accession numbers. Please see http://www.bmj.com/content/340/bmj.c181.long for guidelines on how to de-identify and prepare clinical data for publication. For a list of recommended repositories, please see https://journals.plos.org/plosone/s/recommended-repositories. You also have the option of uploading the data as Supporting Information files, but we would recommend depositing data directly to a data repository if possible. Please update your Data Availability statement in the submission form accordingly. 3. When completing the data availability statement of the submission form, you indicated that you will make your data available on acceptance. We strongly recommend all authors decide on a data sharing plan before acceptance, as the process can be lengthy and hold up publication timelines. Please note that, though access restrictions are acceptable now, your entire data will need to be made freely accessible if your manuscript is accepted for publication. This policy applies to all data except where public deposition would breach compliance with the protocol approved by your research ethics board. If you are unable to adhere to our open data policy, please kindly revise your statement to explain your reasoning and we will seek the editor's input on an exemption. Please be assured that, once you have provided your new statement, the assessment of your exemption will not hold up the peer review process.

Reviewers' comments:

Reviewer's Responses to Questions

**Comments to the Author**

1. Is the manuscript technically sound, and do the data support the conclusions?

Reviewer #1: Yes

Reviewer #2: Partly

2. Has the statistical analysis been performed appropriately and rigorously? 

Reviewer #1: Yes

Reviewer #2: Yes

3. Have the authors made all data underlying the findings in their manuscript fully available?

Reviewer #1: Yes

Reviewer #2: Yes

4. Is the manuscript presented in an intelligible fashion and written in standard English?

Reviewer #1: Yes

Reviewer #2: Yes

5. Review Comments to the Author

Reviewer #1: This is a large-scale, population-based study primarily exploring the impact of BMI on hepatocellular carcinoma (HCC) risk in patients with chronic liver diseases. While several studies have already shown that obesity and BMI can increase the risk of HCC, this study investigates various chronic liver diseases such as HBV, HCV, ALD, NAFLD (non-alcoholic fatty liver disease), and LC, as well as the effects of age and gender, providing new evidence. However, several points need further discussion.

Major comments:

1. What are the diagnostic criteria for alcoholic liver disease and non-alcoholic fatty liver disease? Are they based on case diagnosis codes?

2. Regarding Figure 3, is there statistical significance when the confidence interval crosses 1? For example, for patients under 65 years old, the odds ratio for NAFLD is 1.2 with a confidence interval of 0.87 to 1.64. Please confirm with a statistician.

3. In the discussion section for the BMI-HCC association stratified by age and gender, it is very complex and may not be easily understood by readers. Please make substantial revisions to this discussion section.

Minor comments:

1. On page 13, item 186, it should be Figure 2. Is this a typos?

2. On page 14, item 192, the title of Figure 3 is incorrect. Please revise it.

3. Please specify whether the definition of obesity based on BMI uses Western standards or Asian standards (BMI >23 or >25)

4. On page 18, item 287: the abbreviation full name of HLD ?

Reviewer #2: In this study, Yi et al. investigated the association between BMI and HCC risk in patients with chronic liver disease. They found that higher BMIs were associated with increased HCC risks in patients with different etiologic chronic liver disease. Although the results showed clinically important, several points need be critically addressed.

Specific comments

1. Current evidence showed that several viral factors before antiviral therapies are associated with the risk of HBV and HCV related HCC. In addition, antiviral therapies decrease the risk of HBV and HCV related HCC. The major limitations of this study include the lack of baseline HBV and HCV related viral factors and the data of antiviral therapy.

2. The authors should describe the time point of baseline data collection.

3. Body weight may change during the follow-up period. Only baseline BMI was used for analysis, the impact of changes of BMI on HCC risk cannot be estimated.

4. As above mention, the authors should consider using BMI as a time-dependent covariate to analyze the impact of the dynamic change of BMI on HCC risk.

5. Regarding to the exclusion criteria, only 5062 patients with two or more chronic liver disease were excluded. Currently, the prevalence of NAFLD is so high, but the proportion of HBV or HCV combined with fatty liver is relatively low in this study. Please explain why?

6. The diagnosis of patients was by the ICD-10 codes. How to confirm the coding is correct? For example, whether different doctors have the same diagnostic criteria for ALD.

6. PLOS authors have the option to publish the peer review history of their article (what does this mean?). If published, this will include your full peer review and any attached files.

Reviewer #1: No

Reviewer #2: No

---

## [Author Response · Author response to Decision Letter 0]

19 Sep 2024

We would like to express our sincere appreciation for carefully reviewing our paper and providing valuable feedback. Please find our point-by-point responses to your specific comments. Thank you.

---

## [Decision Letter · Decision Letter 1]

8 Oct 2024

PONE-D-24-15224R1Impact of high body mass index on hepatocellular carcinoma risk in chronic liver disease: A population-based prospective cohort studyPLOS ONE

Dear Dr. Yi,

Thank you for submitting your manuscript to PLOS ONE. After careful consideration, we feel that it has merit but does not fully meet PLOS ONE’s publication criteria as it currently stands. Therefore, we invite you to submit a revised version of the manuscript that addresses the points raised during the review process.

We look forward to receiving your revised manuscript.

Kind regards,

Ashwani Singal

Academic Editor

PLOS ONE

Journal Requirements:

Reviewers' comments:

Reviewer's Responses to Questions

**Comments to the Author**

1. If the authors have adequately addressed your comments raised in a previous round of review and you feel that this manuscript is now acceptable for publication, you may indicate that here to bypass the “Comments to the Author” section, enter your conflict of interest statement in the “Confidential to Editor” section, and submit your "Accept" recommendation.

Reviewer #1: All comments have been addressed

Reviewer #2: All comments have been addressed

2. Is the manuscript technically sound, and do the data support the conclusions?

Reviewer #1: Yes

Reviewer #2: Yes

3. Has the statistical analysis been performed appropriately and rigorously? 

Reviewer #1: Yes

Reviewer #2: Yes

4. Have the authors made all data underlying the findings in their manuscript fully available?

Reviewer #1: Yes

Reviewer #2: Yes

5. Is the manuscript presented in an intelligible fashion and written in standard English?

Reviewer #1: Yes

Reviewer #2: (No Response)

6. Review Comments to the Author

Reviewer #1: In the paper assessing the impact of BMI on the risk of hepatocellular carcinoma across various chronic liver diseases including HBV, HCV, ALD, NAFLD and LC, the authors have answered all the questions I previously raised comprehensively. After reviewing the clarifications, I have no additional questions this time. The detailed explanations provided valuable insight, and I appreciate the thoroughness of your work. The concerns about dual publication, research or publication ethics were not found.

Reviewer #2: This revised manuscript is improved and all previous comments were responded on point-to-point basis. However, the authors cannot analyze the impact of the dynamic change of BMI on HCC risk. The authors should provide more evidence to support that a single BMI measurement can determine the correlation with HCC.

7. PLOS authors have the option to publish the peer review history of their article (what does this mean?). If published, this will include your full peer review and any attached files.

Reviewer #1: No

Reviewer #2: No

---

## [Author Response · Author response to Decision Letter 1]

22 Oct 2024

Thank you for your thoughtful feedback. Please find our point-by-point responses to your specific comments.

---

## [Decision Letter · Decision Letter 2]

9 Dec 2024

Impact of high body mass index on hepatocellular carcinoma risk in chronic liver disease: A population-based prospective cohort study

PONE-D-24-15224R2

Dear Dr. Yi,

We’re pleased to inform you that your manuscript has been judged scientifically suitable for publication and will be formally accepted for publication once it meets all outstanding technical requirements.

Kind regards,

Sona Frankova, M.D., PhD.

Academic Editor

PLOS ONE

Additional Editor Comments (optional):

Reviewers' comments:

Reviewer's Responses to Questions

**Comments to the Author**

1. If the authors have adequately addressed your comments raised in a previous round of review and you feel that this manuscript is now acceptable for publication, you may indicate that here to bypass the “Comments to the Author” section, enter your conflict of interest statement in the “Confidential to Editor” section, and submit your "Accept" recommendation.

Reviewer #1: All comments have been addressed

Reviewer #2: All comments have been addressed

2. Is the manuscript technically sound, and do the data support the conclusions?

Reviewer #1: Yes

Reviewer #2: Yes

3. Has the statistical analysis been performed appropriately and rigorously? 

Reviewer #1: Yes

Reviewer #2: Yes

4. Have the authors made all data underlying the findings in their manuscript fully available?

Reviewer #1: Yes

Reviewer #2: Yes

5. Is the manuscript presented in an intelligible fashion and written in standard English?

Reviewer #1: Yes

Reviewer #2: Yes

6. Review Comments to the Author

Reviewer #1: The revisions effectively address the raised concerns, and the updated manuscript demonstrates clarity and improved scientific rigor. The explanations provided for methodology and data interpretation are convincing, and the adjustments to the discussion strengthen the overall narrative. I appreciate your careful attention to detail and your dedication to enhancing the manuscript’s quality.

Reviewer #2: This revised manuscript is improved and the previous comment was fully

responded. I have no more comments.

7. PLOS authors have the option to publish the peer review history of their article (what does this mean?). If published, this will include your full peer review and any attached files.

Reviewer #1: No

Reviewer #2: No

---

## [Editor Report · Acceptance letter]

10 Jan 2025

PONE-D-24-15224R2 

PLOS ONE

Dear Dr. Yi, 

I'm pleased to inform you that your manuscript has been deemed suitable for publication in PLOS ONE. Congratulations! Your manuscript is now being handed over to our production team.

Kind regards, 

on behalf of

Dr. Sona Frankova 

Academic Editor

PLOS ONE